# Constructed Risk Prognosis Model Associated with Disulfidptosis lncRNAs in HCC

**DOI:** 10.3390/ijms242417626

**Published:** 2023-12-18

**Authors:** Xiao Jia, Yiqi Wang, Yang Yang, Yueyue Fu, Yijin Liu

**Affiliations:** State Key Laboratory of Medicinal Chemical Biology, College of Pharmacy and Tianjin Key Laboratory of Molecular Drug Research, Nankai University, Tianjin 300000, China; 1120210643@mail.nankai.edu.cn (X.J.); 2120221665@mail.nankai.edu.cn (Y.W.); 2120221670@mail.nankai.edu.cn (Y.Y.); 2120211398@mail.nankai.edu.cn (Y.F.)

**Keywords:** hepatocellular carcinoma, risk prognostic model, long-stranded noncoding RNAs, disulfidptosis, PLBD1-AS1, MKLN1-AS

## Abstract

Disulfidptosis is a novel cell death mode in which the accumulation of disulfide bonds in tumor cells leads to cell disintegration and death. Long-stranded noncoding RNAs (LncRNAs) are aberrantly expressed in hepatocellular carcinoma (HCC) and have been reported to carry significant potential as a biomarker for HCC prognosis. However, lncRNA studies with disulfidptosis in hepatocellular carcinoma have rarely been reported. Therefore, this study aimed to construct a risk prognostic model based on the disulfidptosis-related lncRNA and investigate the mechanisms associated with disulfidptosis in hepatocellular carcinoma. The clinical and transcriptional information of 424 HCC patients was downloaded from The Cancer Genome Atlas (TCGA) and divided into test and validation sets. Furthermore, 1668 lncRNAs associated with disulfidptosis were identified using Pearson correlation. Six lncRNA constructs were finally identified for the risk prognostic model using one-way Cox proportional hazards (COX), multifactorial COX, and lasso regression. Kaplan–Meier (KM) analysis, principal component analysis, receiver operating characteristic curve (ROC), C-index, and column-line plot results confirmed that the constructed model was an independent prognostic factor. Based on the disulfidptosis risk score, risk groups were identified as potential predictors of immune cell infiltration, drug sensitivity, and immunotherapy responsiveness. Finally, we confirmed that phospholipase B domain containing 1 antisense RNA 1 (PLBD1-AS1) and muskelin 1 antisense RNA (MKLN1-AS) were highly expressed in hepatocellular carcinoma and might be potential biomarkers in HCC by KM analysis and quantitative real-time PCR (RT-qPCR). This study demonstrated that lncRNA related to disulfidptosis could serve as a biomarker to predict prognosis and treatment targets for HCC.

## 1. Introduction

Hepatocellular carcinoma (HCC) is the sixth most common cancer in the world and the second leading cause of death from cancer [1]. Nearly 850,000 patients develop liver cancer each year and, by 2030, the number of deaths due to HCC is expected to reach 1 million per year worldwide [2,3]. Hepatocellular carcinoma (HCC) is the most common form of liver cancer, accounting for about 90% of all primary tumor cases [2]. The main risk factors for the development of HCC include cirrhosis, hepatitis B virus (HBV) infection, hepatitis C virus (HCV) infection, alcohol abuse, and metabolic syndrome [4]. The treatment of HCC consists of five main modalities, including surgical resection, liver transplantation, chemoembolization, and use of the multi-kinase inhibitor sorafenib [4]. Among these, surgical resection is the most common treatment for patients with hepatocellular carcinoma; however, approximately 70% of patients will develop recurrent HCC after surgical resection, so the prognosis of such patients is typically poor [5]. Therefore, the identification of reliable and specific therapeutic targets and prognostic models is important for improving the early diagnosis rate of hepatocellular carcinoma, evaluating the prognosis, and developing new treatment strategies.

Disulfide is a relatively stable product that is mainly responsible for maintaining the stability of protein structures [6]. A recent study has proposed a novel type of cell death, disulfidptosis, which differs from apoptosis, autophagy, ferroptosis, and cuproptosis. It refers to the fact that, in glucose-starved tumor cells, solute carrier family 7-member 11 (SLC7A11) overexpression leads to a massive depletion of nicotinamide adenine dinucleotide phosphate (NADPH), which in turn triggers an abnormal accumulation of disulfide bonds [7]. This accumulation disrupts the normal binding of disulfide bonds between cytoskeletal proteins, leading to conformational changes in cytoskeletal proteins and rapid tumor cell death [8]. Disulfidptosis may yield a new field of tumor therapy in the future, but its role in HCC remains unclear.

Long-stranded noncoding RNAs (lncRNAs) are a heterogeneous set of nonprotein coding transcripts that are more than 200 nucleotides in length [9]. A large number of lncRNAs are aberrantly expressed in hepatocellular carcinoma compared with normal liver tissue, and they may play critical roles in hepatocarcinogenesis and metastasis [10,11]. For example, retrotransposon Gag like 1 (HuR1) interacts with tumor protein p53 (p53) and represses the transcriptional regulation of downstream genes such as cyclin kinase inhibitor (p21) and bcl2-associated X (Bax) in Hepg2 cells, promoting the proliferation of hepatocellular carcinoma cells [12]. Epigenetically induced myc interacting lncRNA 1 (EPIC1) promotes the development of the hepatocellular carcinoma cell cycle by interacting with myc proto-oncogene, bHLH transcription factor (MYC), and overexpression of EPIC1 correlates with a poor prognosis in hepatocellular carcinoma patients [13]. The lncRNA WD repeat containing antisense to TP53 (WRAP53) is an independent prognostic biomarker that predicts a high rate of recurrence in patients with HCC [14]. However, few studies have focused on the use of lncRNAs associated with disulfidptosis genes for predicting prognosis in HCC patients.

Therefore, this study aims to identify the lncRNAs associated with disulfidptosis in HCC and predict the survival of patients by constructing a risk prognostic model. Our study demonstrates the role of disulfidptosis patterns in HCC for prognosis prediction and provides novel insights for the diagnosis, treatment, and prognosis of HCC.

## 2. Results

### 2.1. Construction of the Prognostic Model Associated with Disulfidptosis lncRNAs for HCC

To explore sulfur-disulfide-related lncRNAs in hepatocellular carcinoma, we screened a total of 1668 lncRNAs associated with sulfur disulfide in hepatocellular carcinoma based on human relevance. Table 1 lists the top 100 lncRNAs associated with disulfidptosis genes. We also plotted the Sanger diagram (Figure 1A). The results suggest that disulfidptosis plays an important and complex role in hepatocellular carcinoma. Subsequently, 249 prognostic lncRNAs associated with disulfidptosis were obtained by using one-way Cox proportional hazards (COX) regression analysis (Figure 1B). Based on the lasso regression, we finally identified six more effective and predictive lncRNAs associated with disulfidptosis: phospholipase B domain containing 1 antisense RNA 1 (PLBD1-AS1), growth-arrest-associated lncRNA 1 (GASAL1), AC128687.2, muskelin 1 antisense RNA (MKLN1-AS), AC026412.3, and long intergenic nonprotein coding RNA 1269 (LINC01269), respectively. We drew a clustered heat map and constructed a risk prognosis model (Figure 1C–E). In addition, we plotted the positive and negative correlations between the expression of these six lncRNAs and the expression of the ten disulfide death genes; the results are shown in Appendix A.

### 2.2. Assessment and Validation of the Risk Prognostic Models for Hepatocellular Carcinoma Associated with Disulfidptosis lncRNAs

To evaluate and validate the risk prognostic model based on lncRNAs associated with disulfidptosis in hepatocellular carcinoma, we randomly divided The Cancer Genome Atlas (TCGA) into a training set and a validation set in a ratio of 6:4. We plotted the distribution of risk scores, the overall survival rate of patients, and the gene expression profiles of the six lncRNAs in the risk prognostic model for the whole TCGA dataset (Figure 2A,D,G,J), as well as the TCGA training (Figure 2B,E,H,K) and validation sets (Figure 2C,F,I,L), as shown in Figure 2. The results indicate that patients had significantly lower survival and higher mortality with increasing risk scores. We further evaluated the predictive ability of the risk prognostic model across the entire dataset, training set, and validation set through Kaplan–Meier (KM) analysis, and the results indicated that the prognosis of the high-risk population was significantly lower than that for the patients with low-risk scores in all three groups (Figure 2G–I).

To further validate the grouping ability of our constructed risk prognostic model, we explored the ability of the four groups—the whole gene, the disulfide gene, all lncRNAs associated with the disulfide gene, and the six screened lncRNAs—to discriminate between high- and low-risk populations using 2D and 3D principal component analysis (PCA). The results of the two-dimensional PCA analysis demonstrated that the first two principal components of all the genes used to differentiate between high- and low-risk populations were 3.15% and 1.73%, respectively, for a total of 4.88%. The proportions of the first two components of the disulfidptosis genes were 19.96% and 13.19%, respectively, for a total of 33.15%. The proportions of lncRNAs associated with disulfidptosis were 3.59% and 1.97%, for a total of 5.56%. The proportions of the six lncRNAs in our constructed risk prognostic model were 25.61%, 19.2%, 15.06%, 14.44%, 13.22%, and 12.45%, respectively; the first two principal components accounted for 44.81%, and the six lncRNAs accounted for 99.98% in total. Based on the above results, we can conclude that our constructed risk model had the largest variance contribution, and the six lncRNAs basically included all the populations. From the 2D and 3D PCA analysis plots (Figure 3, Appendix A), it can be more intuitively seen that our constructed risk prognostic model was capable of significantly differentiating between high- and low-risk populations among the patients. In summary, our results show that the first three sets of models had very poor ability to distinguish between high- and low-risk populations, whereas the proposed risk diagnostic model could very accurately classify the populations into high- and low-risk populations (Figure 3A–D).

To further validate whether our constructed risk prognostic model could serve as an independent prognostic factor and its predictive performance, we also conducted single-factor regression, multifactor regression, receiver operating characteristic curve (ROC), and C-index analyses for evaluation. The results of the single-factor regression analysis suggested that our risk prognostic model had a hazard ratio (HR) of 1.120 (1.074–1.167), with a very significant and statistically significant *p* < 0.001, such that the risk model might be an independent prognostic factor (Figure 4A). The results of the multifactorial regression analysis corroborated the previous results that the proposed risk model was independent of other clinical factors and was an independent prognostic factor for predicting patients with hepatocellular carcinoma (HR value of 1.142, *p* < 0.001) (Figure 4B). The results of the ROC analysis reinforced the fact that the risk prognostic model was independent of other clinical factors, and it evaluated patients with hepatocellular carcinoma at 1 year with a predicted area under curve (AUC) value of 0.751, those at 3 years with a predicted AUC value of 0.643, and those at 5 years with a predicted AUC value of 0.660. Both the abovementioned results and the C-index results confirmed that the risk prognostic model constructed based on disulfidptosis-associated lncRNAs was independent of the clinical factors and might be considered as an independent prognostic factor for the prediction of patients with hepatocellular carcinoma (Figure 4C–E).

### 2.3. Clinical Subgroup Validation of the Risk Prognostic Model for Hepatocellular Carcinoma Associated with Disulfidptosis lncRNAs

TheKM method is currently the most commonly used method for survival analysis, proposed by Kaplan and Meier. We performed KM analysis in different clinical subgroups to further validate our constructed risk prognostic model and explore the correlation between this risk score and clinical characteristics. The clinical correlation results suggested that there was no statistically significant correlation between the risk score and age and gender, and the correlation between three clinical characteristics—namely T-stage, tumor grade, and pathological staging of the tumor—and the risk model was statistically significant (*p* < 0.05) (Figure 5A–C). In addition, we further validated the predictive performance of our constructed risk model by clinical grouping. The results suggested that the *p*-values of the KM analysis results for patients whose pathological grading was located in the Stage I–IV grading and whose tumor grading was classified in the M0-, N0-, and T2-stage subgroups were all less than 0.05, further confirming that the risk prognostic model constructed based on disulfidptosis-associated lncRNAs can be considered as an independent prognostic factor for the prediction of hepatocellular carcinoma (Figure 5D–G).

### 2.4. Functional Enrichment Analysis of the Risk Prognostic Model Associated with Disulfidptosis lncRNAs in Hepatocellular Carcinoma

To further explore the functional signaling pathways that may be enriched in patients in the risk prognostic model, we performed Gene Ontology (GO), Kyoto Encyclopedia of Genes and Genomes (KEGG), and gene set enrichment analysis (GSEA) analyses. The GO results suggested that biological process (BP) was mainly associated with nuclear division, mitotic nuclear division, extracellular matrix organization, extracellular structure organization, and external encapsulating structure organization; molecular function (MF) was mainly associated with receptor ligand activity, endopeptidase activity, tubulin binding, extracellular matrix structural constituent, glycosaminoglycan binding, and microtubule binding; and cellular component (CC) was mainly related to the collagen-containing extracellular matrix, apical part of cell, apical plasma membrane, spindle, microtubule, and basal part of cell (Figure 6A,B). The KEGG results suggested that these genes were mainly enriched in the phosphatidylinositol3-kinase (PI3K)-protein kinase B (Akt) signaling pathway, cell cycle, cytokine–cytokine receptor interaction, proteoglycans in cancer, focal adhesion, protein digestion and absorption, motor proteins, the extracellular matrix (ECM)-receptor interaction, Hippo signaling pathway, and other pathways (Figure 6C,D). The GSEA results suggested that basal cell carcinoma, cytokine–cytokine receptor interaction, ECM receptor interaction, and other pathways were active in the high-risk population; fatty acid degradation, Clycine serine, and threonine metabolism were active in the low-risk population; and proteasome and primary bile acid biosynthesis pathways were generally active (Figure 6E,F).

### 2.5. Immunosignature Characterization of the Risk Prognostic Model Associated with Disulfidptosis lncRNAs in Hepatocellular Carcinoma

To further explore the immune characteristics of the population in our constructed risk prognostic model, we mapped Figure 7A,B with ssGSEA. The results indicated that there was a significant difference in the tumor immune infiltration characteristics of the high- and low-risk populations. In addition, we also mapped the immune profiles of individual genes to further validate our risk prognostic model (Appendix A). In addition, we also explored the immune cell scoring and tumor microenvironment between the two groups, the results of which confirmed that the tumor microenvironment (TME) scores of the patients in the high-risk group were lower than those in the low-risk group and that the patients in the high-risk group had higher tumor purity; furthermore, the relative immune cell content in these patients was lower, and the prognosis was poorer (Figure 7C,D).

The tumor mutational burden (TMB) score is commonly used to assess the immunotherapy response of patients in risk prognostic models. The TMB results suggested that the high-risk group had higher TMB scores and may have better immunotherapy outcomes (Figure 8A). We also analyzed the results of the 15 genes with the highest mutation frequencies in the high- and low-risk groups (Figure 8B,C). The results suggested that the five most frequently mutated genes in the high-risk population were TP53, catenin beta 1 (CTNNB1), titin (TTN), mucin 16, cell-surface-associated (MUC16), and piccolo presynaptic cytomatrix protein (PCLO). Tumor immune dysfunction and rejection (TIDE) is commonly used to assess the likelihood of tumor immune escape in patients, and our results demonstrate that the outcomes for the high-risk population could be improved through immunotherapy (Figure 8D).

### 2.6. Potential Therapeutic Agents in Hepatocellular Carcinoma

To explore drug sensitivity in the high- and low-risk populations, as well as potential drugs for treating patients with hepatocellular carcinoma, we used the pRRophetic algorithm to predict drug sensitivity in these patients. The results showed that the high-risk group may be more sensitive to 90 drugs. These drugs may be considered as potential drugs for the treatment of liver cancer patients; specifically, these included ABT737, Afatinib, AIG-5198, Alpelisib, AMG-319, AZD4547, and Cisplatin (Figure 9).

### 2.7. Construction of Line Graphs for the Risk Prognostic Modelling

To revalidate our constructed risk prognostic model, we constructed column-line plots for a particular patient with hepatocellular carcinoma and evaluated the model in terms of the patient’s 1-, 3-, and 5-year survival rates. Our column-line calibration results suggested that the risk prognostic model constructed based on disulfidptosis-associated lncRNAs had good predictive performance (Figure 10).

### 2.8. Exploring Potential Prognostic Therapeutic Targets in Hepatocellular Carcinoma

To further evaluate and validate the potential of the six lncRNAs employed in constructing the risk prognostic model as tumor markers, we explored the prognostic value of these six lncRNAs using KM analysis (Figure 11). The results suggested that two lncRNAs—PLDB1-AS1 and MKLNS1-AS—were the most statistically significant (*p* < 0.0001) (Figure 11A,B). Therefore, we selected these two lncRNAs to further explore their expression in hepatocellular carcinoma. The RT-qPCR results suggested that they were both differentially expressed in hepatocellular carcinoma cell lines with statistical significance compared to cells in normal liver tissue (LO2) (Figure 11C,D). This also reaffirmed that our risk prognostic model constructed based on disulfidptosis-associated lncRNAs might be an independent prognostic factor for predicting hepatocellular carcinoma patients.

## 3. Discussion

The recurrence rate and therapeutic efficacy of HCC are important issues in the field of tumor therapy. Despite the availability of several therapeutic approaches, including the application of systemic therapy and immunotherapy, the prognosis is generally poor for HCC patients. The development of sequencing technology has provided an opportunity to screen specific biomarkers associated with HCC progression. Disulfidptosis—that is, the accumulation of disulfide bonds in tumor cells leading to cell disintegration and death—has aroused extensive research interest among scholars as a novel mode of cell death [7,15]. However, the roles of lncRNAs associated with disulfidptosis in hepatocellular carcinoma have not yet been clarified. In this study, we explored the lncRNAs associated with disulfidptosis in HCC and constructed a novel risk prognostic model based on lncRNAs to provide a more accurate tool for prognostic assessment of hepatocellular carcinoma patients.

To explore the critical role of disulfidptosis in cancer and the interactions between disulfide apoptosis and lncRNAs, we identified and validated a set of key lncRNAs associated with disulfidptosis in HCC, including PLBD1-AS1, GASAL1, AC128687.2, MKLN1-AS, AC026412.3, and LINC01269. These lncRNAs may be involved in the regulation of disulfidptosis through multiple pathways. For example, the RT-qPCR results in our study suggested that PLBD1 was highly expressed in hepatocellular carcinoma cells compared to normal hepatocytes, consistent with the findings of Luo [16]. PLBD1-AS1 may promote the development of hepatocellular carcinoma by activating autophagy by affecting the TP53- and CHMP4B-mediated DNA damage response [17]. GASAL1 was overexpressed in Hepg2 and Huh7. Knockdown of this expression may inhibit the proliferation and migration ability of hepatocellular carcinoma cells by regulating miR-193b-5p, which, in turn, affects ubiquitin specific peptidase 10 (USP10) [18]. Compared with LO2, AC026412.3 was also highly expressed in a variety of hepatocellular carcinoma cells (Hepg2, Huh7, and Hep3B), which may be associated with poor prognosis in hepatocellular carcinoma patients [19]. Elevated MKLN1-AS has been demonstrated to be one of the causes of poor prognosis in hepatocellular carcinoma patients, which is also consistent with our findings [20]. Through RT-qPCR, we further confirmed by RT-qPCR that MKLN1-AS expression was significantly elevated in hepatocellular carcinoma cells compared with normal hepatocytes. Both in vivo and in vitro experiments confirmed that MKLN1 may promote the development of hepatocellular carcinoma by affecting yes1-associated transcriptional regulator (YAP1) [21]. The discovery of these lncRNAs broadens our understanding of the pathogenesis of HCC. AC128687.2 and LINC01269 have rarely been reported in the study of hepatocellular carcinoma and deserve to be followed up through in-depth experimental investigations.

Functional enrichment analysis revealed multiple biological processes and signaling pathways in which these six lncRNAs may be involved, including the PI3K-Akt signaling pathway and cell cycle regulation, which are closely related to the development of liver cancer. These results provide important clues to reveal the potential mechanisms of these lncRNAs in liver cancer development, which may contribute to future functional experimental studies and provide new therapeutic targets for the treatment of liver cancer.

We subsequently integrated the six key disulfidptosis-related lncRNAs into a risk prognostic model. In the testing and validation sets, the model successfully categorized hepatocellular carcinoma patients into high- and low-risk groups, demonstrating its excellent predictive performance regarding patient prognosis. More importantly, the risk model constructed based on the six disulfidptosis-associated lncRNAs was an independent prognostic factor for HCC. Further analyses indicated significant correlations between the risk scores and clinical characteristics of hepatocellular carcinoma patients, including T-stage, tumor grade, and pathological staging. This suggests that the risk prognostic models constructed based on these lncRNAs have strong clinical correlations and are expected to provide personalized survival risk assessments in different subgroups of hepatocellular carcinoma patients, which provides strong support for the development and adjustment of therapeutic regimens.

Given the important role of the TME in tumorigenesis and progression, interactions between cancer cells and immune cells regulate all aspects of tumor development. Thus, disulfidptosis-associated lncRNA mechanisms may influence tumor progression through immune-related pathways. Through immune infiltration analysis, we observed significant differences in tumor immune infiltration between the high- and low-risk groups. The high-risk group may have a poorer tumor immune environment, but their higher TMB scores may make them more sensitive to immunotherapy. This suggests that the proposed risk prognostic model can also provide important information for immunotherapy decision making, including potential guidance for the individualized treatment of hepatocellular carcinoma patients. On this basis, a drug sensitivity analysis showed that drugs such as ABT737, Afatinib, and Alpelisib may be potential choices for the treatment of hepatocellular carcinoma patients in the high-risk group. As a B-cell lymphoma-2 (Bcl-2) anti-apoptotic protein inhibitor, ABT737 has been shown to have potent anti-tumor effects against several types of tumors, including HCC [22]. Studies in ovarian cancer patients have shown that ABT737 can enhance the sensitivity of cancer cells to Cisplatin by modulating mitochondrial autophagy and glucose metabolism [23,24]. Afatinib is mainly used for the treatment of advanced non-small-cell lung cancer, but recent studies have shown that Afatinib can enhance the sensitivity of HCC tumors through signal transducer and activator of transcription 3 (STAT3)/ CD274 molecule (PD-L1) pathway cells with PD-L1 expression, and its combination with anti-PD1 therapy significantly increased the immunotherapeutic efficacy of HCC [25]. Alpelisib is a selective phosphatidylinositol-4,5-bisphosphate 3-kinase catalytic subunit alpha (PIK3CA) inhibitor, which is effective in treating PIK3CA-mutated HCC by inhibiting the mitogen-activated kinase-like protein (MAPK) and AKT cascade response. In addition, Alpelisib presented synergistic efficacy in PIK3CA mutant HCC in combination with mechanistic target of rapamycin kinase (mTOR) or cyclin-dependent kinase (CDK) 4/6 inhibitors [26].

Although this study has achieved important progress in revealing the mechanisms underlying disulfidptosis and lncRNA regulation in HCC, some limitations remain. First, we identified coexpressed lncRNAs associated with disulfide bond death and constructed a prognostic model based on six lncRNAs, but further exploration of the association between core lncRNAs and HCC prognosis as well as validation of the generalization of the prognostic model in clinical samples are still needed. Second, our study focused on the regulatory mechanism of lncRNAs, and more detailed functional and mechanistic studies are still required. Furthermore, while we found that MKLN1-AS was associated with expression of the disulfide death genes (GYS1, NCKAP1, and OXSM), how the regulation between them works remains to be studied. For this, we plan to construct knockdown and overexpression lncRNA cell lines in a follow-up study to explore and verify the obtained results in depth at the cellular and animal levels through molecular biology experiments such as RT-qPCR, Western blot, luciferase reporter gene, and so on. This is a direction of our future research that needs to be deepened. In addition, while we explored the immune profile of the population associated with the risk prognostic model and the relationships between immune infiltration and individual genes, more experiments are required to demonstrate how these lncRNAs are related to immune infiltration (e.g., with respect to M0 macrophages). In addition, the relationship between disulfidptosis and the therapeutic strategy and prognostic assessment of HCC needs to studied in more depth. In the future, we intend to conduct more experimental validation and clinical studies in order to further clarify the biological functions and potential mechanisms of these lncRNAs in the context of hepatocellular carcinoma.

## 4. Materials and Methods

### 4.1. Data Acquisition

The TCGA project was jointly launched by the National Cancer Institute and the National Human Genome Research Institute in 2006, aiming to use genome sequencing and bioinformatics to analyze gene mutations responsible for cancer. RNA-seq data, clinical information, and mutation information of 424 HCC patients (including 374 liver cancer and 50 normal liver tissues) were downloaded from TCGA on 12 July 2023 (https://portal.gdc.cancer.gov/repository). The clinical data included age, gender, histologic grade, pathological stage, pathological T-stage, pathological M-stage, pathological N-stage, survival time, and survival status. Practical extraction and report language (perl) is a computer programming language that can process text files and retrieve relevant information from them. Transcripts per million (TPM) is a metric commonly used in RNA sequencing studies. We used perl scripts to collate the transcriptomics data downloaded from TCGA into TPM expression matrices for all genes. Next, we used perl scripts on the collated gene expression matrix to differentiate between mRNAs and lncRNAs in order to obtain separate mRNA and lncRNA expression matrices [27,28,29].

### 4.2. Identification of lncRNAs Associated with Disulfidptosis in HCC

We adopted the same research criteria as in several previous studies. A total of 10 disulfidptosis genes were obtained from the previously published literature: GYS1, NDUFS1, OXSM, LRPRC, NDUFA11, NUBPL, NCKAP1, RPN1, SLC3A2, and SLC7A11 [30,31,32,33]. Subsequently, we performed Pearson correlation analysis of the disulfidptosis genes and lncRNA genes to obtain lncRNAs associated with disulfidptosis in hepatocellular carcinoma using a threshold of |Pearson| ≥ 0.3 and *p* < 0. 001.

### 4.3. Construction of a Predictive Model for HCC Based on Disulfidptosis-Related lncRNAs

We used the survival package in R 4.3.1 to conduct one-way Cox proportional risk regression in order to identify lncRNAs associated with disulfidptosis in hepatocellular carcinoma (*p* < 0.05). Subsequently, lasso regression using the glmnet package in R was conducted for further screening, allowing us to obtain lncRNAs significantly different from overall survival period (OS), adjusting the parameter setting to lambda. min [34]. Finally, multifactorial Cox regression was computed using the survival package in R, allowing for identification of the lncRNAs used to construct the disulfidptosis risk prognostic model. We divided the samples in the TCGA dataset according to a ratio of 6:4, which were randomized into training and test sets. We assessed the risk scores of liver cancer patients according to the following formula [35]:$$risk\;scores=\sum_{i=1}^{n}coef(mfrlncRNA\;i)\times expr(mfrlncRNA\;i)$$

Patients with hepatocellular carcinoma were categorized into high- or low-risk groups based on the median risk score.

### 4.4. Validation of the Risk Prognostic Models and Construction of Column-Line Diagrams

Principal component analysis is the most widely used data dimensionality reduction algorithm, commonly used to distinguish sample categories. PCA and risk score correlation analysis were conducted to validate the ability of the risk model to distinguish between high- and low-risk groups [36]. Consistency index (C-index) and ROC curves were used to assess the accuracy of the risk model. A Kaplan–Meier analysis was performed between the high- and low-risk groups to assess whether the risk score could serve as an independent predictor of clinical prognosis [37]. Univariate and multivariate Cox regression analyses were carried out to calculate the prognostic value of the prognostic model regarding various clinical characteristics [38]. Finally, calibration curves were calculated using the calibration “rms” package, and prognostic plots were constructed for patients with hepatocellular carcinoma at 1, 3, and 5 years using clinical characteristics (age, sex, grade, T-stage, M-stage, and N-stage) and risk scores to assess the predictive ability of the bisulfite-death-associated lncRNAs-associated risk-based prognostic model. We also constructed prognostic plots of patients with liver cancer at 1, 3, and 5 years.

### 4.5. Immune Infiltration and Functional Analysis

Expression data (ESTIMATE) were utilized to estimate the content of stromal and immune cells in malignant tumor tissues, thus allowing for assessment of tumor purity [39,40]. Immune infiltration results for high- and low-risk patients were assessed by seven algorithms, including TIMER, CIBERSORT, quantTIseq, xCell, MCPcounter, and EPIC [41,42,43]. In addition, we used the Maftool package to analyze and visualize the frequency of mutated genes and common TMB genes in the high- and low-risk groups. The TIDE and TME scores were used to explore whether there existed differences in immune response and tumor microenvironment infiltration between the different groups of patients [44].

### 4.6. Biological Function Enrichment Analysis

To explore potential functional pathways or differences in the biological function of lncRNAs associated with disulfidptosis in hepatocellular carcinoma patients, we performed functional enrichment analyses using the cluster profiles in the GSEA [45]. Pathways with *p* < 0.05 were considered statistically significant.

### 4.7. Drug Sensitivity Analysis

To explore potential drugs that may be effective in liver cancer patients, we also assessed drug sensitivity in liver cancer patients using the pRRophetic algorithm, with the threshold set at *p* < 0.001 [46]. The lower the IC_50_ value, the higher the sensitivity to the drug, and the better the guidance regarding the patient’s clinical use of the drug.

### 4.8. Cell Culture

Three hepatocellular carcinoma cell lines (Huh7, Hepg2) and normal liver tissue cells (LO2) were purchased from the cell bank of the Shanghai Institutes for Biological Sciences, Chinese Academy of Sciences (Shanghai, China). The Huh7, Hepg2, and LO2 cells were cultured in Dulbecco’s modified eagle’s medium (DMEM) medium with 10% fetal bovine serum (FBS). Both survived in cultures incubated at 37 °C in a 37 °C incubator containing 5% CO_2_. All media were changed once every two days.

### 4.9. Quantitative Reverse Transcription PCR (RT-qPCR)

Total RNA was extracted from normal liver tissue cells (LO2) and two types of hepatocellular carcinoma cells (Hepg2 and Huh7) according to the instructions of the Trizol kit. The total RNA extracted was reverse transcribed into cDNA using a reverse transcription reagent according to the instructions of the reagent vendor. Quantitative Real-time PCR were performed using a SYBR-GREEN kit (Yeasen Biotechnology, Shanghai, China). The qPCR upstream and downstream primers were as follows: GAPDH(F): ACCCAGAAGACTGTGGATGG; GAPDH(R): TTCAGCTCAGGGATGACCTT; PLBD1-AS1(F): GTGGATTCCATCCTAGAGGCTGTG; PLBD1-AS1(R): TTCCTGCTTTCTGTCCTTCATTTCAG; MKLN1-AS(F): ACTGGGTCTGAGGTGTAAGC; and MKLN1-AS(R): TGATGACACTGTCCAGGCTT. All results were calculated using the 2^−ΔΔCT^ method after normalizing to GAPDH.

### 4.10. Statistical Analysis

Statistical analysis was performed using the GraphPad Prism 9 software. A *t*-test was conducted to assess the difference between the two groups of data. All data are expressed as mean ± SEM. *p* < 0.05 was considered statistically significant.

## 5. Conclusions

In this study, we constructed a risk prognostic model based on six disulfidptosis-related lncRNAs. KM, PCA, ROC, C-index, and column-line plot analyses confirmed that the proposed risk prognostic model may be useful for independent prognostic determination. The KM and RT-qPCR analysis results confirmed that PLBD1-AS1 and MKLN1-AS may be potential biomarkers for hepatocellular carcinoma.

## Figures and Tables

**Figure 1 ijms-24-17626-f001:**
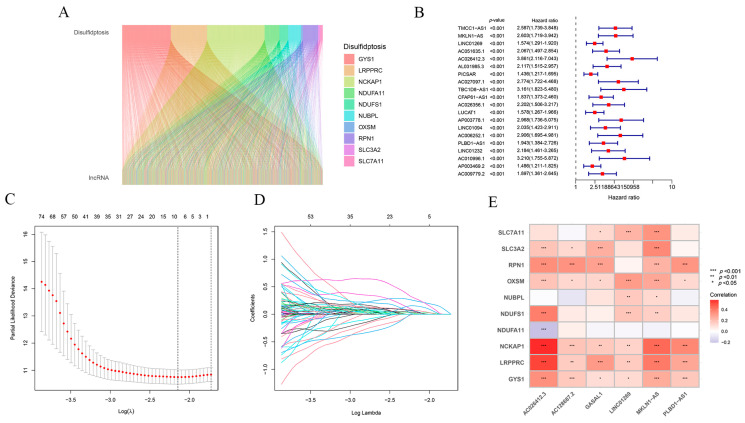
Construction of the prognostic model for lncRNAs associated with disulfidptosis in hepatocellular carcinoma. (**A**) Pearson correlation identification of the coexpression of disulfidptosis genes in hepatocellular carcinoma derived from the lncRNAs Sanger map. (**B**) One-way regression analysis of lncRNAs associated with disulfidptosis in hepatocellular carcinoma. (**C**,**D**) Lasso analysis of lncRNAs associated with disulfidptosis in hepatocellular carcinoma. Among them, the two dashed lines in (**C**) indicate two special λ values; one is lambda. Min and the other lambda. 1se. (**D**) reflects the importance of each variable, with different colored lines representing different variables as the penalty term increases λ as the quantity increases. (**E**) Clustering heatmap of 10 disulfidptosis genes associated with 6 lncRNAs analyzed.

**Figure 2 ijms-24-17626-f002:**
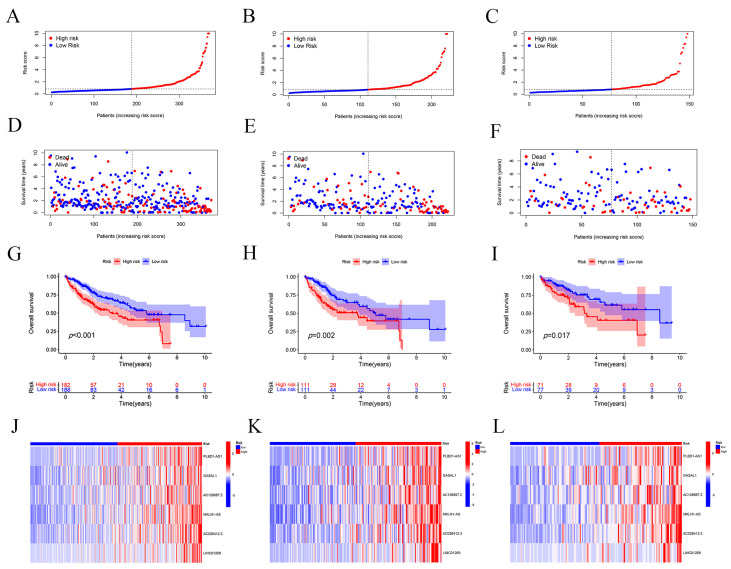
Grouping and assessment of the risk prognostic models. (**A**) Scatterplot of risk scores for patients in the entire TCGA dataset. (**B**) Scatterplot of risk scores for patients in the TCGA training set. (**C**) Scatterplot of risk scores for patients in the TCGA validation set. (**D**) Scatterplot of survival status for patients in the TCGA entire dataset. (**E**) Scatterplot of survival status for patients in the TCGA training set. (**F**) Scatterplot of survival status of patients in the TCGA validation set. (**G**) Kaplan–Meier survival plot for patients in the entire TCGA dataset. (**H**) Kaplan–Meier survival plot for patients in the TCGA training set. Among them, the dashed line in the middle of subfigures (**A**–**F**) is also known as the median value, mainly used to distinguish between high- and low-risk populations. The left side of the median is low risk, represented by blue, and the right side is high risk, represented by red. (**I**) Kaplan–Meier survival plot of patients in the TCGA validation set. (**J**) Cluster analysis plot of the risk prognostic model for these 6 lncRNAs in patients of the TCGA entire dataset. (**K**) Cluster analysis plot of the risk prognostic model for these 6 lncRNAs in patients in the TCGA training set. (**L**) Cluster analysis plot of the risk prognostic model for these 6 lncRNAs in patients in the TCGA validation set.

**Figure 3 ijms-24-17626-f003:**
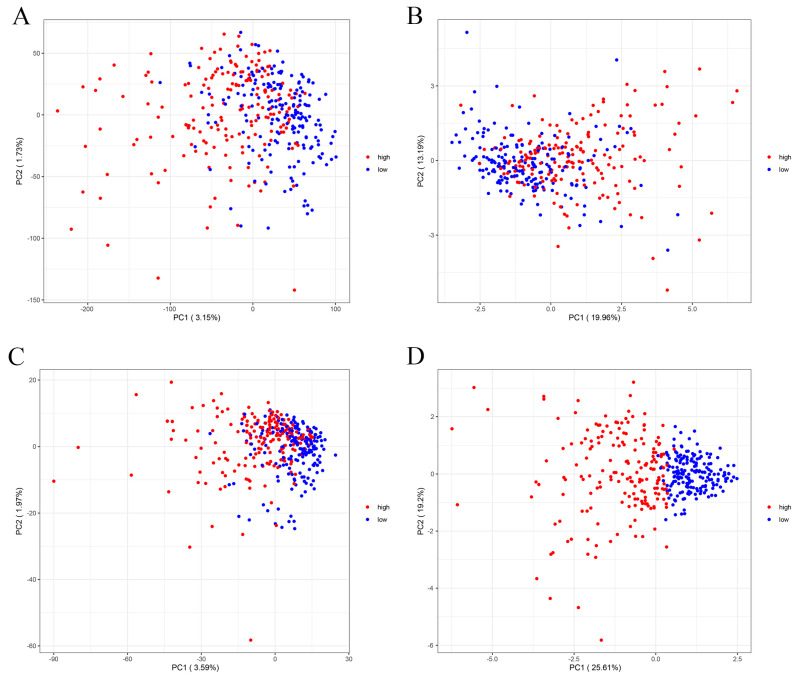
The ability of PCA to assess the grouping of risk prognostic models. (**A**) The ability of PCA to analyze whole genes to distinguish between high- and low-risk populations. The first two principal components of all genes used to differentiate between high- and low-risk populations were 3.15 percent and 1.73 percent. (**B**) The ability of PCA to analyze disulfidptosis genes to distinguish between high- and low-risk populations. The first two components of the disulfidptosis genes were 19.96% and 13.19%. (**C**) PCA analysis of the ability of disulfidptosis-associated lncRNAs to distinguish between high- and low-risk populations. The first two components of lncRNAs associated with disulfidptosis were 3.59% and 1.97%. (**D**) PCA analysis of the ability of our constructed risk prognostic model to distinguish between high- and low-risk populations. The first two components of these six lncRNAs in our constructed risk prognostic model were 25.61% and 19.2%.

**Figure 4 ijms-24-17626-f004:**
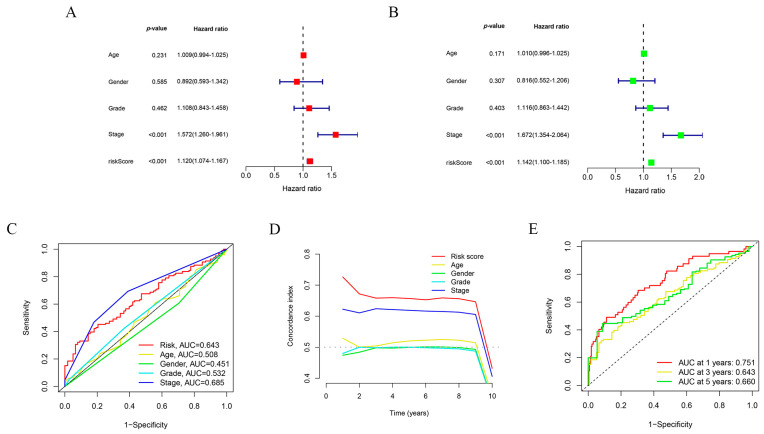
Validation of the predictive performance of the risk prognostic models. (**A**) One-way regression analysis to validate the predictive performance of the risk prognostic model constructed with lncRNAs associated with disulfidptosis in hepatocellular carcinoma. (**B**) Multifactor regression analysis validating the predictive performance of the risk prognostic models constructed with lncRNAs associated with disulfidptosis in hepatocellular carcinoma. (**C**) ROC analysis to validate the predictive performance of risk prognostic models constructed by lncRNAs associated with disulfidptosis in hepatocellular carcinoma. (**D**) C-index validation of the predictive performance of risk prognostic models constructed with lncRNAs associated with disulfidptosis in hepatocellular carcinoma. The dotted line represents a C-index value of 0.5, and the model on the dotted line indicates a good predictability of the risk prognostic model. (**E**) ROC analysis to validate the predictive performance of risk prognostic models constructed by lncRNAs associated with disulfidptosis in hepatocellular carcinoma.

**Figure 5 ijms-24-17626-f005:**
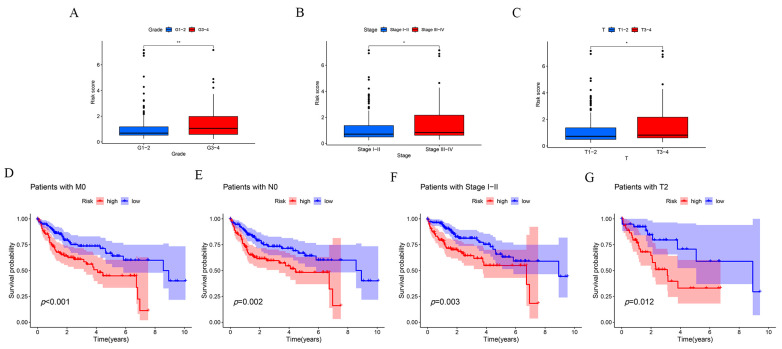
Clinical subgroup validation of the risk prognostic models and correlation analysis of clinical features. (**A**) Tumor grading characteristics of the risk prognostic model. (**B**) Pathology grading features of the risk prognostic model. (**C**) Tumor T-grading characteristics of the risk prognostic model. (**D**) Kaplan–Meier survival curve analysis of M0-staged patients. (**E**) Kaplan–Meier survival curve analysis of N0-staged patients. (**F**) Kaplan–Meier survival curve analysis of patients with pathologic grading Stage I-IV. (**G**) Kaplan–Meier survival curve analysis graph of patients with T-stage. * *p*-value < 0.05, ** *p*-value < 0.01.

**Figure 6 ijms-24-17626-f006:**
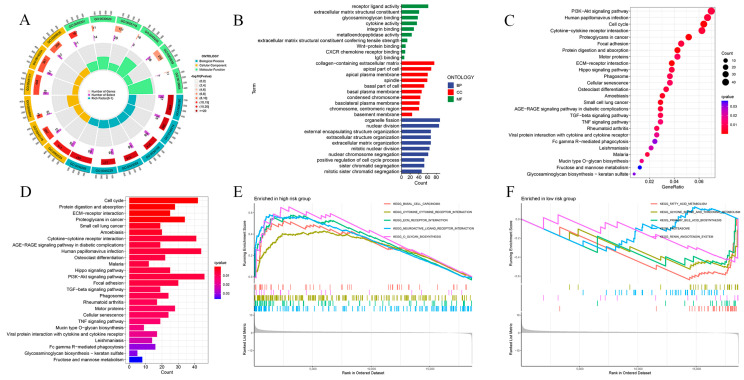
Signaling pathways that may be enriched in high- and low-risk populations. (**A**) Circular graph of GO pathway enrichment results. (**B**) Bar graph of GO pathway enrichment results. (**C**) Bubble diagram of KEGG pathway enrichment results. (**D**) Bar graph of KEGG pathway enrichment results. (**E**) GSEA enrichment results of a high-risk population. (**F**) GSEA enrichment results of low-risk population.

**Figure 7 ijms-24-17626-f007:**
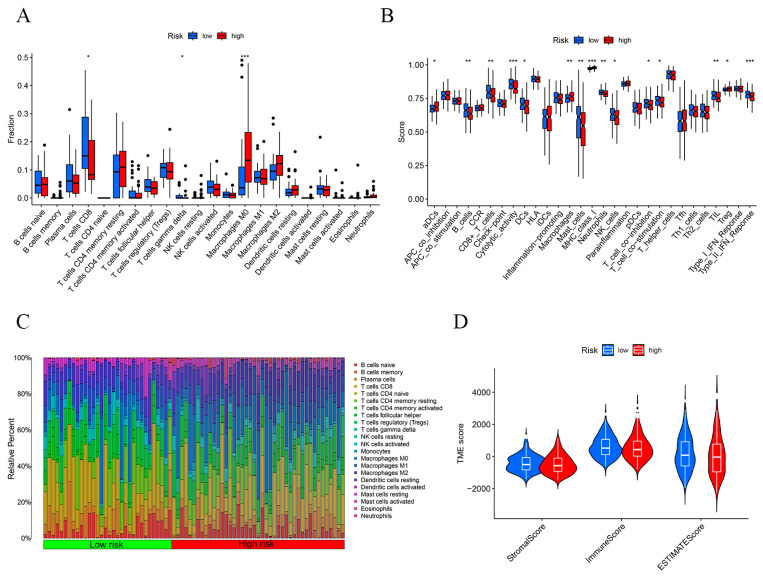
Immunological characterization of the risk prognostic model. (**A**) Immunofunctional analysis. (**B**) Immune score analysis. (**C**) Immune cell percentage in high- and low-risk groups. (**D**) Tumor microenvironment analysis of both high- and low-risk groups. * *p*-value < 0.05, ** *p*-value < 0.01, *** *p*-value < 0.001.

**Figure 8 ijms-24-17626-f008:**
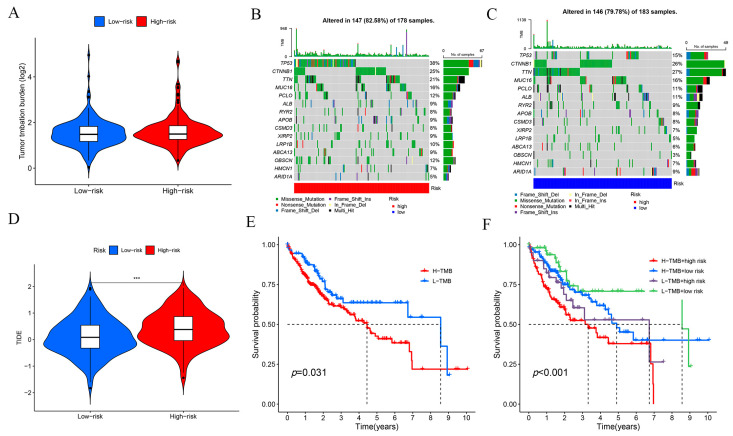
Immunosignature analysis of the risk prognostic models. (**A**) TMB scoring for both high- and low-risk groups. (**B**) Waterfall plot of the top 15 mutated genes common to the high-risk group. (**C**) Waterfall plot of the top 15 mutated genes common to the low-risk group. (**D**) TIDE analysis of the high- and low-risk groups. (**E**) KM survival analysis of the two groups with high and low TMB scores. (**F**) KM survival analysis of the four groups with high TMB score and high risk, high TMB score and low risk, low TMB score and high risk, and low TMB score and low risk. *** *p*-value < 0.001.

**Figure 9 ijms-24-17626-f009:**
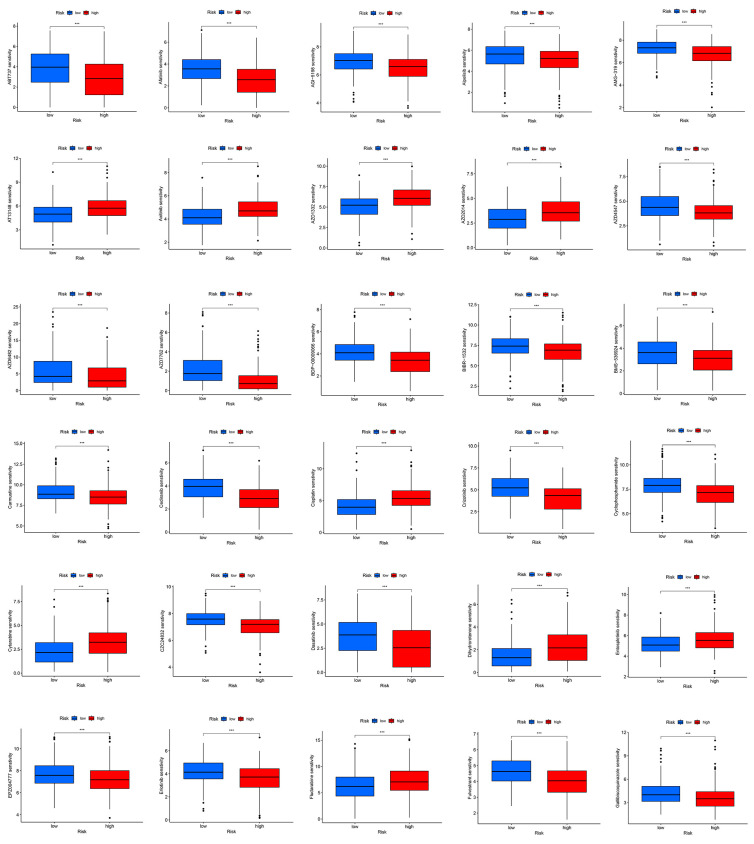
Drug susceptibility analysis of the risk prognostic model. *** *p*-value < 0.001.

**Figure 10 ijms-24-17626-f010:**
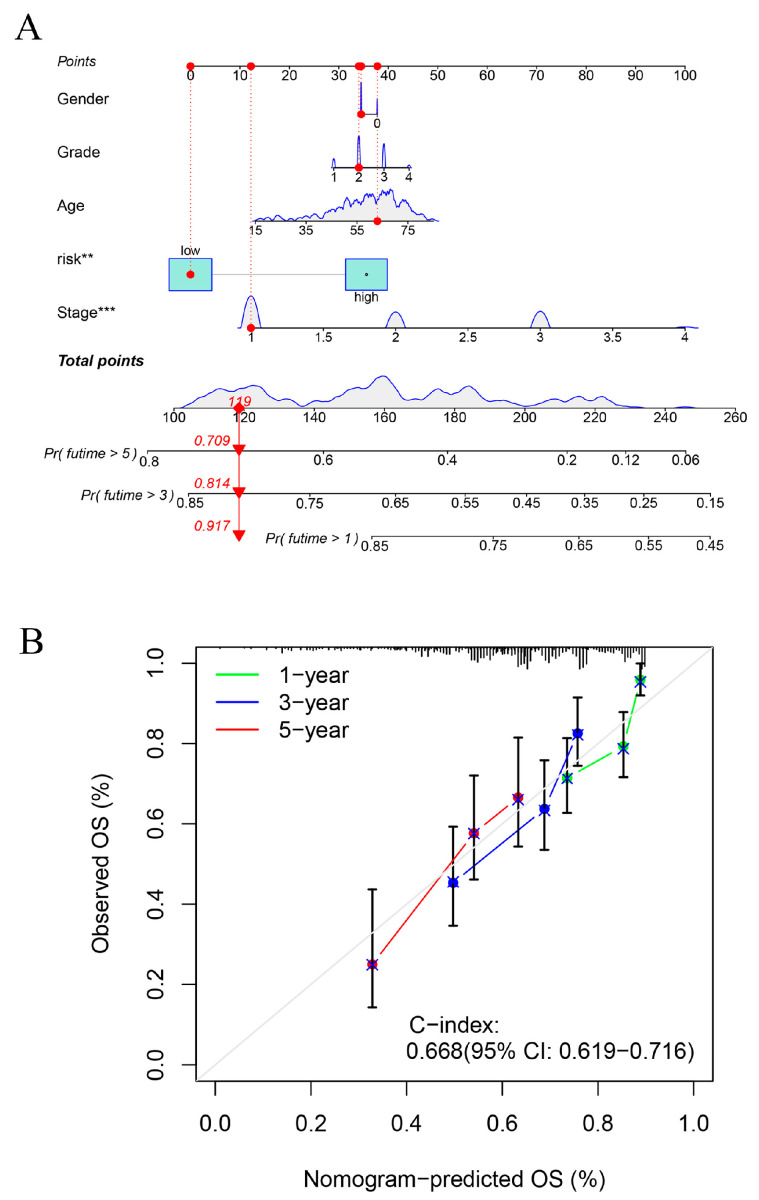
Construction and assessment of the risk prognostic column-line diagrams. (**A**) Prognostic column-line diagram for a particular patient. (**B**) Column-line calibration chart. ** *p*-value < 0.01, *** *p*-value < 0.001.

**Figure 11 ijms-24-17626-f011:**
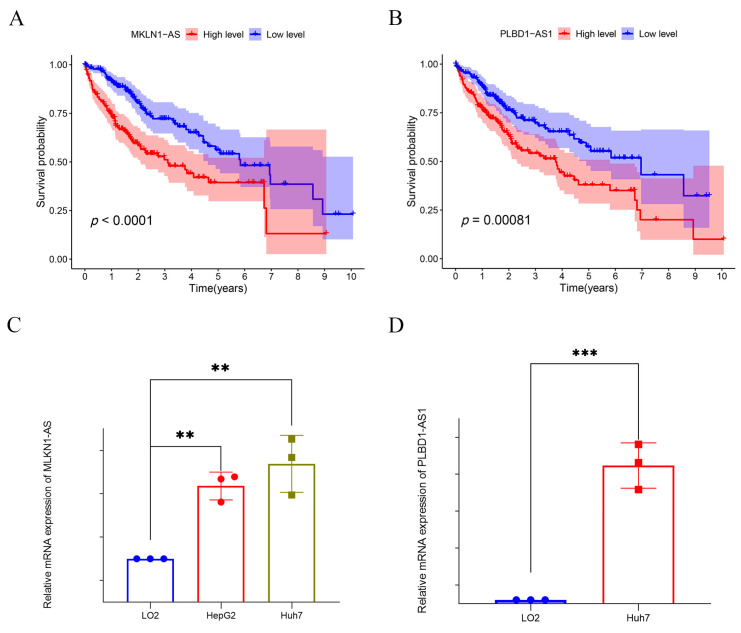
Prognostic value of the six lncRNAs modelled. (**A**,**B**) KM survival analysis of PLDB1-AS1 and MKLNS1-AS1. (**C**,**D**) PLDB1-AS1 and MKLNS1-AS1 were differentially expressed in normal hepatocytes and hepatocellular carcinoma cell lines. ** *p*-value < 0.01, *** *p*-value < 0.001.

**Table 1 ijms-24-17626-t001:** Top 100 lncRNAs associated with disulfidptosis gene in hepatocellular carcinoma.

Disulfidptosis	lncRNA	Cor	*p*-Value	Regulation
NCKAP1	FGD5-AS1	0.755540944	2.54 × 10^−70^	positive
NCKAP1	AC092614.1	0.700809965	1.58 × 10^−56^	positive
NCKAP1	Z68871.1	0.697550131	8.32 × 10^−56^	positive
NCKAP1	NORAD	0.697398002	8.99 × 10^−56^	positive
NCKAP1	AC005670.3	0.695765397	2.05 × 10^−55^	positive
NUBPL	AC008549.1	0.688460193	7.66 × 10^−54^	positive
NDUFA11	SNHG25	0.68845343	7.68 × 10^−54^	positive
NCKAP1	EBLN3P	0.674755577	5.15 × 10^−51^	positive
NCKAP1	AC004596.1	0.670828403	3.12 × 10^−50^	positive
NCKAP1	FAM111A-DT	0.663170449	9.68 × 10^−49^	positive
NCKAP1	AC073046.1	0.658844394	6.44 × 10^−48^	positive
NCKAP1	NRAV	0.658106969	8.87 × 10^−48^	positive
NCKAP1	FBXL19-AS1	0.656078366	2.13 × 10^−47^	positive
NCKAP1	AC112220.2	0.642313785	6.81 × 10^−45^	positive
NCKAP1	NNT-AS1	0.641857367	8.21 × 10^−45^	positive
NCKAP1	LINC01560	0.641104761	1.11 × 10^−44^	positive
NCKAP1	AC007406.4	0.638614637	3.05 × 10^−44^	positive
NUBPL	AC010501.2	0.638601193	3.07 × 10^−44^	positive
LRPPRC	ZNF337-AS1	0.637634279	4.53 × 10^−44^	positive
NCKAP1	LINC02035	0.636883618	6.12 × 10^−44^	positive
LRPPRC	MCM3AP-AS1	0.635449648	1.08 × 10^−43^	positive
NUBPL	TPRG1-AS1	0.632600955	3.35 × 10^−43^	positive
NCKAP1	AC108463.2	0.629554128	1.10 × 10^−42^	positive
NCKAP1	AC010834.3	0.62874749	1.51 × 10^−42^	positive
LRPPRC	FGD5-AS1	0.628682804	1.55 × 10^−42^	positive
NCKAP1	NRSN2-AS1	0.627176632	2.78 × 10^−42^	positive
NCKAP1	LINC01278	0.626164716	4.10 × 10^−42^	positive
NCKAP1	AL157392.3	0.624970416	6.49 × 10^−42^	positive
NCKAP1	STARD7-AS1	0.623197317	1.28 × 10^−41^	positive
NCKAP1	OIP5-AS1	0.623017826	1.37 × 10^−41^	positive
LRPPRC	AC092614.1	0.621999965	2.01 × 10^−41^	positive
GYS1	NRSN2-AS1	0.621082258	2.85 × 10^−41^	positive
NCKAP1	CTBP1-DT	0.619754599	4.70 × 10^−41^	positive
NUBPL	LINC01124	0.619531729	5.11 × 10^−41^	positive
NCKAP1	AC011462.5	0.616702842	1.47 × 10^−40^	positive
NCKAP1	AC107027.3	0.613268075	5.23 × 10^−40^	positive
NCKAP1	RNF213-AS1	0.612672379	6.51 × 10^−40^	positive
NCKAP1	SMARCA5-AS1	0.61225062	7.59 × 10^−40^	positive
NCKAP1	LINC00205	0.609943603	1.76 × 10^−39^	positive
NUBPL	GCC2-AS1	0.609278159	2.24 × 10^−39^	positive
NCKAP1	ZEB1-AS1	0.60887102	2.60 × 10^−39^	positive
LRPPRC	AC005670.3	0.608849702	2.62 × 10^−39^	positive
NCKAP1	MCM3AP-AS1	0.608419664	3.06 × 10^−39^	positive
NDUFS1	OIP5-AS1	0.604600905	1.20 × 10^−38^	positive
NCKAP1	HCG18	0.603891569	1.55 × 10^−38^	positive
NCKAP1	ZNF32-AS2	0.600052829	5.99 × 10^−38^	positive
LRPPRC	AC016747.1	0.599723623	6.73 × 10^−38^	positive
NCKAP1	LINC01521	0.599402932	7.52 × 10^−38^	positive
NCKAP1	AC097448.1	0.599076405	8.43 × 10^−38^	positive
NCKAP1	AC006008.1	0.598792563	9.31 × 10^−38^	positive
NCKAP1	AC107959.3	0.598162326	1.16 × 10^−37^	positive
NDUFA11	TP53TG1	0.596806014	1.86 × 10^−37^	positive
NCKAP1	SBF2-AS1	0.596758995	1.89 × 10^−37^	positive
NCKAP1	WAC-AS1	0.59524344	3.19 × 10^−37^	positive
NCKAP1	ZNF337-AS1	0.594297989	4.41 × 10^−37^	positive
NDUFA11	AC108673.3	0.594176279	4.60 × 10^−37^	positive
NCKAP1	AP001469.2	0.593039476	6.79 × 10^−37^	positive
GYS1	STARD7-AS1	0.590300693	1.73 × 10^−36^	positive
LRPPRC	HCG18	0.588923912	2.75 × 10^−36^	positive
LRPPRC	AL133243.2	0.588483387	3.19 × 10^−36^	positive
NDUFS1	PAXIP1-AS2	0.587674426	4.18 × 10^−36^	positive
LRPPRC	AC114763.1	0.585416633	8.90 × 10^−36^	positive
NCKAP1	AC073254.1	0.58450427	1.20 × 10^−35^	positive
NUBPL	AL132800.1	0.584445842	1.23 × 10^−35^	positive
GYS1	AC090409.1	0.584364815	1.26 × 10^−35^	positive
NCKAP1	AP003392.1	0.584330402	1.28 × 10^−35^	positive
GYS1	AC005261.1	0.583205211	1.85 × 10^−35^	positive
NCKAP1	WARS2-AS1	0.582826349	2.10 × 10^−35^	positive
LRPPRC	EBLN3P	0.580224203	4.94 × 10^−35^	positive
GYS1	AL353748.3	0.579802946	5.67 × 10^−35^	positive
LRPPRC	AC073046.1	0.578759756	7.96 × 10^−35^	positive
LRPPRC	SNHG16	0.577307032	1.28 × 10^−34^	positive
GYS1	WDFY3-AS2	0.577261915	1.29 × 10^−34^	positive
NCKAP1	AC026412.3	0.577055681	1.38 × 10^−34^	positive
NCKAP1	EIF3J-DT	0.575128924	2.58 × 10^−34^	positive
SLC7A11	AC016717.2	0.574785212	2.88 × 10^−34^	positive
NCKAP1	NIPBL-DT	0.573977047	3.73 × 10^−34^	positive
NCKAP1	PAXIP1-AS2	0.572860551	5.33 × 10^−34^	positive
LRPPRC	AC007406.4	0.57199109	7.02 × 10^−34^	positive
NDUFA11	SNHG9	0.570739778	1.04 × 10^−33^	positive
NCKAP1	AC026979.4	0.569894433	1.36 × 10^−33^	positive
NCKAP1	AC016705.2	0.56982153	1.40 × 10^−33^	positive
NDUFS1	AC135050.5	0.569725317	1.44 × 10^−33^	positive
NCKAP1	AC005034.5	0.569677444	1.46 × 10^−33^	positive
GYS1	LINC01772	0.569075281	1.77 × 10^−33^	positive
NCKAP1	WDFY3-AS2	0.56880377	1.92 × 10^−33^	positive
LRPPRC	NORAD	0.568364614	2.21 × 10^−33^	positive
NCKAP1	AC025171.2	0.567681518	2.74 × 10^−33^	positive
NCKAP1	AC114763.1	0.566813614	3.59 × 10^−33^	positive
LRPPRC	CTBP1-DT	0.566805009	3.60 × 10^−33^	positive
NCKAP1	AL499602.1	0.566769896	3.64 × 10^−33^	positive
NCKAP1	LINC00342	0.566560491	3.89 × 10^−33^	positive
NCKAP1	AL133243.2	0.565705344	5.07 × 10^−33^	positive
NCKAP1	AC120114.1	0.565556097	5.31 × 10^−33^	positive
LRPPRC	AC073254.1	0.562259089	1.47 × 10^−32^	positive
GYS1	WARS2-AS1	0.561505107	1.86 × 10^−32^	positive
NCKAP1	LINC00265	0.560909226	2.23 × 10^−32^	positive
GYS1	AP002761.3	0.560771952	2.33 × 10^−32^	positive
NCKAP1	NPTN-IT1	0.560266675	2.71 × 10^−32^	positive
NCKAP1	TRAPPC12-AS1	0.559882531	3.05 × 10^−32^	positive

Note: instructions of the prognostic model for lncRNAs associated with disulfidptosis in hepatocellular carcinoma (NCKAP1), NUBP iron–sulfur cluster assembly factor, mitochondrial (NUBPL); NADH: ubiquinone oxidoreductase subunit A11 (NDUFA11), glycogen synthase 1 (GYS1), 3-oxoacyl-ACP synthase, mitochondrial (OXSM), leucine-rich pentatricopeptide repeat containing (LRPRC), ribophorin I (RPN1), and solute carrier family 3 member 2 (SLC3A2).

## Data Availability

Please do not hesitate to contact us if you need the raw data and original code of the original article.

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
