# Peer review of "Constructed Risk Prognosis Model Associated with Disulfidptosis lncRNAs in HCC"

_ijms, 2023, doi:10.3390/ijms242417626_

Round 1

Reviewer 1 Report

Comments and Suggestions for Authors

In this manuscript, Jia et. al. have explored the prognostic significance of lncRNA related to disulfidptosis. This is an interesting manuscript but there are several areas that need to be improved.

  1. The authors should expand on their lncRNA extraction and normalization procedures.
  2. The authors should consider adding a Kaplan-Meier plot with 95% confidence intervals to their manuscript. Kaplan-Meier plots are a useful way to display survival data, and including confidence intervals can provide additional information about the precision of the estimates.
  3. The authors need to perform principal component analysis (PCA) in two dimensions and report the results, including the variance explained by each principal component and relevant statistical analysis.
  4. The figures in the manuscript should adhere to a consistent convention. For instance, the bar plots and the visualization of statistical significance should be consistent across all figures. Furthermore, the font size, color, line width and other factors should be uniform to ensure that the figures are uniform in their appearance. By following a uniform convention, the figures will be more aesthetically pleasing and easier to interpret.
  5. The authors need to compare all the figures for statistical analysis, for example Figure 7(D).
  6. In this study, TPM data was used, which is a processed count and not a useful metric for comparison across samples. Along with TPM, RPKM, and FPKM, this can lead to spurious results. The authors need to provide more detail on the reasons for their use of processed counts and include references to support their procedure. “Because of the nature of the quantification measures and embedded implicit normalization process, TPM, RPKM, and FPKM expression levels are suitable for the comparison of RNA transcript expression within a single sample. However, none of these measures can be used universally for cross-sample comparisons and downstream analyses such as determining differentially expressed genes between two or more biological states.”
  • Refer: Zhao, Yingdong, Ming-Chung Li, Mariam M. Konaté, Li Chen, Biswajit Das, Chris Karlovich, P. Mickey Williams, Yvonne A. Evrard, James H. Doroshow, and Lisa M. McShane. "TPM, FPKM, or normalized counts? A comparative study of quantification measures for the analysis of RNA-seq data from the NCI patient-derived models repository." Journal of translational medicine 19, no. 1 (2021): 1-15.
  • Zhao, Shanrong, Zhan Ye, and Robert Stanton. "Misuse of RPKM or TPM normalization when comparing across samples and sequencing protocols." Rna 26, no. 8 (2020): 903-909.

  1. The authors need to elaborate on the search criterion, inclusion and exclusion criteria and add references “Ten disulfidptosis genes were obtained from previously published literature. “.
  2. The authors need to cite all the original authors of algorithms that are used in this study.
  3. To ensure scientific reproducibility, it is recommended that the authors make their code available.
  4. The lack of clinical validation using internal cohorts:- (Authors need to use clinical samples and validate the prognostic or higher/lower expression in samples. They can use a variety of techniques, most prominently real-time PCR or protein detection).
Comments on the Quality of English Language

The authors need to thoroughly revise the manuscript for English.

Author Response

Thank you very much for your patient review and revision comments. We very much recognize your point of view. We explained your concerns and recommendations one by one in the following text.Please see the attachment.

Reviewer 2 Report

Comments and Suggestions for Authors

In this study, Jia et al. seek the roles of lncRNAs in HCC. Although topic is tempting, this manuscript lacks critical evidence to support the authors’ conclusion.

·       Multiple previous studies showed candidate lncRNAs associated with HCC and its prognosis, and some studies used TCGA data, so this study lacks novelty. The authors show high MKLN1-AS associated with poor prognosis in Figure 11, but previous studies showed the same result using the same TCGA data (Gao et al. 2020, Chen et al. 2022). The authors use risk scores to predict prognosis in Figure 4, but a study with very similar concepts is already published (Deng et al. 2021). This manuscript does not provide any novel findings to readers.

·       The authors claim that lncRNAs identified in this study are associated with disulfidptosis, but not enough evidence is provided. Figure 1 shows only correlation. There is no experimental evidence to prove that HCC has increased/decreased disulfidptosis, or candidate lncRNAs, such as MKLN1-AS, increase/decrease disulfidptosis in HCC. Correlation or association identified using data from database do not necessarily prove that those lncRNAs mediate disulfidptosis and contribute to poor prognosis. The authors need to show experimental evidence.

·       Same issue for cell fraction in Figure 7. Cell fractions were estimated using data obtained from database, but there is no validation or support with experimental evidence. For example, in Figure 7A, patients with high risk contain higher population of M0 macrophages than patients with low risk. First, this is only computational estimation using database, so experimental evidence is required to support this, such as immunostaining for M0 macrophages in liver sections of low and high risk patients. Second, the association between lncRNAs and elevation of M0 macrophages should be explained. Candidate lncRNAs are associated with macrophage activation? High MKLN1-AS expression inhibits macrophage polarization so they are remaining at M0? The authors should provide more evidence. Overall, this study lacks novelty for readers and experimental evidence to support the conclusion. 

Author Response

(The authors gave the same response as above.)
